# ONLINE SFT FOR LLM REASONING

## ABSTRACT

We present a simple, self-help online supervised finetuning (**OSFT**) method for LLM reasoning. In this paradigm, the model generates its own responses and is immediately finetuned on this self-generated data. OSFT is a highly efficient training strategy for LLM reasoning, as it is *reward-free* and uses just one rollout by default. Experiment results show that OSFT achieves downstream performance on challenging mathematical reasoning tasks comparable to strong reinforcement learning (RL) methods such as GRPO. Our ablation study further demonstrates the efficiency and robustness of OSFT. The major mechanism of OSFT lies in facilitating the model's own existing preference (latent knowledge) learned from pretraining, which leads to reasoning ability improvement. We believe that OSFT offers an efficient and promising alternative to more complex, reward-based training schemes.

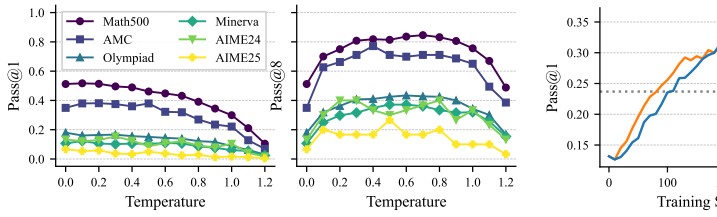 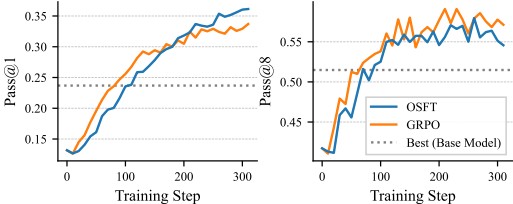

(a) Pass@1 and pass@8 on base model via different validation temperature ($\tau_{eval}$).

(b) Averaged pass@1 and pass@8 across six math benchmarks for models trained by GRPO and OSFT.

Figure 1: Motivation and performance of Online SFT (OSFT) with Qwen2.5-Math-7B as the base model. **(a)** The performance of the untuned base model is sensitive to different evaluation temperatures ($\tau_{eval}$). Pass@1 accuracy (left) peaks at low temperatures, while pass@8 (right) benefits from moderate temperatures. This motivates a training approach that reinforces the model's existing preference learned from pretraining, which leads to OSFT. **(b)** OSFT exhibits downstream performance comparable to the baseline GRPO across six math reasoning benchmarks, and both improve over the base model's best performance. Here, the "Best (Base Model)" horizontal line represents the averaged peak score on these six benchmarks of the untuned base model found by sweeping evaluation temperatures $\tau_{eval}$.

## 1 INTRODUCTION

Large language models (LLMs) (Achiam et al., 2023; Dubey et al., 2024; Yang et al., 2024) have emerged as a promising pathway toward achieving general artificial intelligence (Bubeck et al., 2023). In particular, there has been a surge of interest in training LLMs with Chain-of-Thoughts (CoT) (Wei et al., 2022) reasoning paths for complex mathematical tasks, which has driven notable progress on challenge mathematical benchmarks, as demonstrated by models like OpenAI-o1 and DeepSeek-R1 (Jaech et al., 2024; Guo et al., 2025). Consequently, developing efficient training strategies for reasoning models has become an increasingly important research direction.

**Background and Related Work.** Due to the rapid growth of research on LLM reasoning, we provide a non-exhaustive overview.

To improve LLM reasoning capabilities, reinforcement learning with verifiable reward (RLVR) has become a popular approach (Guo et al., 2025). Many recent works aim to replicate RL's success

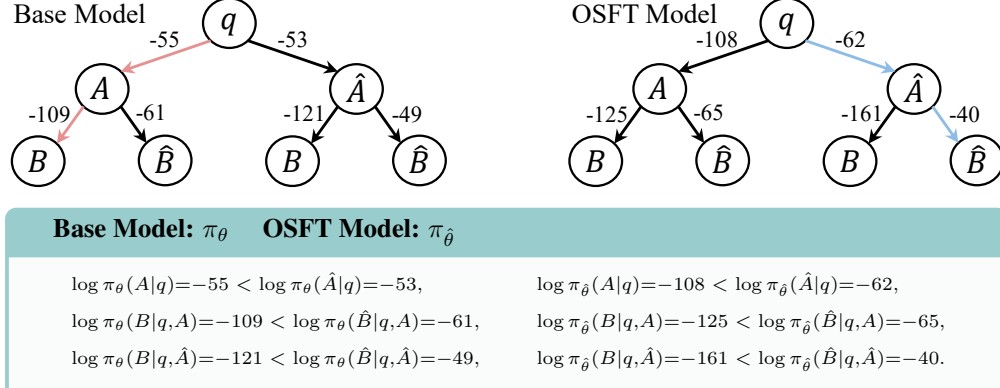

Figure 2: Given the same question $q$, the base model generates the reasoning steps $[A, B]$ with $B$ being the **wrong** response (highlighted in light red, picked from one of all 8 wrong tries), while the OSFT model generates the path $[\hat{A}, \hat{B}]$ with $\hat{B}$ containing the **correct** response (highlighted in light blue). It can be seen that OSFT facilitates the base model's existing preference obtained from pre-training, which largely widens the probability margin between the correct path and its counterparts, leading to correct reasoning. The full question and responses are put in Appendix D. More detailed analysis of this experiment result can be found in Section 4.2.1.

in reasoning; see, e.g., Face (2025); Pan et al. (2025); Luo et al. (2025); Yu et al. (2025); Liu et al. (2025); Zeng et al. (2025); Hu et al. (2025a); Wang et al. (2025b). The core step of RL lies in using a rule-based reward, e.g., the correct answer of the math question, to provide a training signal towards fitting the correctly generated answer by the model itself. Common RL algorithms for reasoning include GRPO (Shao et al., 2024) and PPO (Schulman et al., 2017). There are also many GRPO variants such as DAPO (Yu et al., 2025), Dr. GRPO (Liu et al., 2025), and GSPO (Zheng et al., 2025).

Beyond RL, the standard supervised finetuning (SFT) pipeline on reasoning data distilled from strong reasoning models like R1 can also improve LLM's reasoning ability; see, e.g., Muennighoff et al. (2025); Ye et al. (2025); Li et al. (2025); Wen et al. (2025). The recent work (Guha et al., 2025) thoroughly investigates the reasoning dataset for SFT.

There is also a wide range of work exploring other aspects of LLM reasoning, including the role of RL in incentivizing reasoning (Yue et al., 2025), the 80/20 rule on tokens (Wang et al., 2025a), analyses of models' reasoning processes (Chen et al., 2025), the illusion of thinking (Shojaee et al., 2025), rethinking reward signals (Shao et al., 2025), and entropy-based mechanisms (Cui et al., 2025; Agarwal et al., 2025), improving reasoning ability with one training example (Wang et al., 2025b), survey on efficient reasoning (Sui et al., 2025), among others.

**Main contributions.** In this work, our goal is to provide a simple, efficient, self-help SFT-based algorithm, so that it can be used to improve the reasoning ability of models. Our main contributions can be summarized below.

(C.1) We propose Online SFT (**OSFT**) training method for LLM reasoning. OSFT is an efficient, *reward-free*, and self-help algorithm that improves the model by performing SFT on the self-generated data. One factor contributing to OSFT's efficiency is that it can use only a single rollout, i.e., one sample per prompt, which we adopt as the default setting. To improve the model's reasoning ability, the mechanism of OSFT lies in enhancing the model's existing preference (latent knowledge) obtained from pretraining, as illustrated by Figure 2 and the analysis in Section 4.2.1.

(C.2) We conduct extensive empirical experiments showing that OSFT is a highly effective training strategy. In particular, we show that it achieves downstream performance comparable to the strong baseline GRPO on both math-specific and general-purpose base models across different mathematical reasoning benchmarks. Our ablation study confirms OSFT's efficiency and robustness. In summary, our experiment results indicate that OSFT is a simple yet promising alternative for LLM reasoning.

## 2 PRELIMINARIES

We provide preliminaries abut language modeling and training formulations for incentivizing LLM's reasoning ability in this section.

### 2.1 LANGUAGE MODELS

LLMs are autoregressive models that generate a sequence of tokens given an input. An LLM $\pi_\theta$ with parameters $\theta$ is usually modeled by the transformer architecture (Vaswani et al., 2017). Given a prompt $q$, it generates an output sequence $o = (o_1, o_2, \ldots, o_T)$ through the conditional probability distribution $\pi_\theta(o \mid q)$ in an autoregressive manner, namely,

$$\pi_\theta(o \mid q) = \prod_{t=1}^{T} \pi_\theta(o_t \mid q, o_{<t}), \tag{1}$$

where $o_{<t}$ denotes the tokens preceding $o_t$.

### 2.2 SUPERVISED FINETUNING (SFT)

SFT is a standard technique for adapting a pre-trained model $\pi_\theta$ to specific tasks by training it on a static dataset $\mathcal{D}$ of high-quality prompt-response pairs $(q, o)$. The training objective of SFT is formulated by minimizing the negative log-likelihood loss over $\mathcal{D}$:

$$\mathcal{L}_{\text{SFT}}(\theta; \mathcal{D}) = -\mathbb{E}_{(q,o)\sim\mathcal{D}} \left[ \sum_{t=1}^{|o|} \log \pi_\theta(o_t \mid q, o_{<t}) \right]. \tag{2}$$

SFT is commonly applied for adapting a pre-trained model to follow human instructions or to manage specifc downstream tasks. In terms of incentiving the model's reasoning ability, one can also perform SFT on long CoT data distilled from strong reasoning models, where the response $o$ contains rich reasoning traces; see, e.g., Muennighoff et al. (2025); Ye et al. (2025); Li et al. (2025); Wen et al. (2025); Guha et al. (2025).

### 2.3 REINFORCEMENT LEARNING WITH VERIFIABLE REWARDS

For reasoning tasks where solutions can be programmatically verified, on-policy RL is a popular approach for improving the model's reasoning ability. A popular method in this domain is Group Relative Policy Optimization (GRPO) (Shao et al., 2024). GRPO is to maximize the following clipped surrogate objective derived from Proximal Policy Optimization (PPO):

$$
\begin{aligned}
J(\theta) =\mathbb{E}_{q\sim\mathcal{D}, o\sim\pi_{\text{old}}(\cdot|q)} \bigg[ \sum_{t=1}^{|o|} & \min \bigg( \frac{\pi_\theta(o_t|q, o_{<t})}{\pi_{\text{old}}(o_t|q, o_{<t})} \, \hat{A}(q, o), \\
& \text{clip} \left( \frac{\pi_\theta(o_t|q, o_{<t})}{\pi_{\text{old}}(o_t|q, o_{<t})}, 1 - \epsilon, 1 + \epsilon \right) \hat{A}(q, o) \bigg) \bigg] - \beta \, \mathbb{E}_{q\sim\mathcal{D}} \Big[ D_{\text{KL}}\big(\pi_\theta(\cdot|q)\|\pi_{\text{ref}}(\cdot|q)\big) \Big],
\end{aligned}
\tag{3}
$$

where $\hat{A}(q, o) = \frac{r(q,o) - \bar{r}_q}{\sigma_q}$ is the advantage with $r$ being reward and $D_{\text{KL}}$ is the KL divergence.

## 3 OSFT: ONLINE SUPERVISED FINETUNING

In this section, we introduce the Online Supervised Finetuning (OSFT) method, an iterative, reward-free, self-help algorithm designed to improve LLM's reasoning ability.

### 3.1 MOTIVATION: THE CORRELATION BETWEEN CERTAINTY AND PERFORMANCE

The initial motivation for our work comes from Figure 1a, where we analyze the performance of the base Qwen2.5-Math-7B model across various sampling temperatures. It can be observed that for every mathematical reasoning benchmark, the pass@1 performance is highest in the low-temperature

---

**Algorithm 1** Iterative Online Supervised Finetuning (OSFT)

---

**Input** initial model $\pi_{\theta_{\text{init}}}$; task prompts $\mathcal{D}$; hyperparameters: sampling temperature $\tau_s$, training temperature $\tau_t$, rollouts per prompt $G$

1: model initialization $\pi_\theta \leftarrow \pi_{\theta_{\text{init}}}$
2: **for** step = 1, . . . , M **do**
3:     $\pi_{\text{old}} \leftarrow \pi_\theta$
4:     Sample a batch of questions $\mathcal{D}_b$ from $\mathcal{D}$
5:     Initialize an empty set for the training batch: $\mathcal{D}_{\text{OSFT}} \leftarrow \varnothing$
6:     **for** each question $q \in \mathcal{D}_b$ **do**
7:         Sample $G$ outputs $\{o_i\}_{i=1}^G \sim \pi_{\text{old}}(\cdot \mid q, \tau_s)$
8:         Add the generated pairs $\{(q, o_i)\}_{i=1}^G$ to $\mathcal{D}_{\text{OSFT}}$
9:     **for** OSFT iteration = 1, . . . , $\mu$ **do**
10:         Update the model $\pi_\theta$ by minimizing the OSFT loss (Equation (4)) on $\mathcal{D}_{\text{OSFT}}$.

**Output** $\pi_\theta$

---

region ($\tau_{eval} \leq 0.6$) and degrades significantly as the temperature increases. For pass@8, peak performance is observed in the mid-temperature range ($\tau_{eval} \approx 0.4$–$0.8$). This observation motivates us to enhance the model's certainty about its existing preferences obtained from pretraining, but only to a reasonable extent, as pass@8 (i.e., exploration ability) declines at extremely low temperatures, in order to improve LLM's mathematical reasoning ability.

### 3.2 OSFT: A Simple Method for LLM Reasoning

Motivated by the above observations, we propose **Online SFT (OSFT)** for LLM reasoning. It is a simple, reward-free, self-help algorithm designed to iteratively amplify the model's certainty on its existing preference obtained from the pretraining phase. The core loop, illustrated in Algorithm 1 (see also Figure 13 for a workflow), involves two steps:

1. **Self data generation:** Using the model itself to sample outputs with a low sampling temperature $\tau_s$.
2. **SFT:** The model is then updated by performing SFT with a temperature $\tau_t$ on these self-generated data.

This online, self-help process lends itself to a natural comparison with RL algorithms like GRPO, but their core dynamics are fundamentally different. As outlined in Section 2.3, GRPO's update is driven by a reward-based advantage, targeting at the importance sampling ratio $\pi_\theta(o_t|q, o_{<t})/\pi_{\text{old}}(o_t|q, o_{<t})$. For this ratio to be a valid measure of policy/model change, the sampling distribution $\pi_{\text{old}}$ and training distribution $\pi_\theta$ are usually set to be comparable, as is done in the VERL platform (Sheng et al., 2025). OSFT, in sharp contrast, is a reward-free algorithm. It has no advantage term nor the importance sampling ratio. OSFT simply optimizes the SFT's negative log-likelihood loss over the data generated by the old model, and hence the rationale for sampling and training consistence (such as temperature coupling) is no longer required. Indeed, we will illustrate below that the sampling and training distributions have to be different in OSFT, as otherwise the learning signal will be null.

This major difference allows us to decouple the sampling and training distributions for specialized roles. We formalize this by defining the OSFT loss with distinct sampling and training temperatures:

$$\mathcal{L}_{\text{OSFT}} = -\mathbb{E}_{q \sim \mathcal{D}, o \sim \pi_{\text{old}}(\cdot|q;\tau_s)} \left[\log \pi_\theta(o \mid q; \tau_t)\right] \tag{4}$$

While in principle $\tau_t$ can be tuned, we find that a standard, non-aggressive setting of $\tau_t = 1$ is sufficient for stable learning, which we adopt in our main experiments.

### 3.3 Discussion on Decoupled Temperature Dynamics in OSFT

The decoupled temperature setting of OSFT is not an empirical accident but a direct consequence of one-step gradient dynamics. We now discuss the interplay between $\tau_s$ and $\tau_t = 1$ to show that the condition

$$\tau_s < \tau_t \tag{5}$$

is necessary for stable learning in OSFT by checking its one-step gradient.

**The Unstable Regime ($\tau_s \geq \tau_t$).** In this regime, OSFT fails because the learning signal is either directionless or actively destructive.

*Case 1: Coupled Temperatures ($\tau_t = \tau_s = \tau$).* When temperatures are identical, the expected parameter update is directionless. This is a direct consequence of the score-function identity (see Appendix E.1):

$$\mathbb{E}_{o \sim \pi_\theta(\cdot|q;\tau)}[\nabla_\theta \log \pi_\theta(o \mid q; \tau)] = 0. \tag{6}$$

This means $\mathbb{E}[\theta_{\text{new}}] = \theta_{\text{old}}$. Our experiment results also verify this observation; see Figure 7.

*Case 2: Inverted Temperatures ($\tau_s > \tau_t$).* This scenario creates a destructive mismatch. The sampling process, governed by a higher $\tau_s > 1$, is more stochastic and generates a diverse set of outputs that have even worse performance; see Figure 1a. Therefore, the model is trained to behave more randomly, leading to worse reasoning ability.

**The Stable Regime ($\tau_s < \tau_t$).** For one token response case, we have (see Appendix E.2)

$$\mathbb{E}_{o \sim \pi_\theta(\cdot|q;\tau_s)}[\nabla_\theta \log \pi_\theta(o \mid q; \tau_t)] = \frac{1}{\tau_t} \cdot J_\theta \left[ p_{\tau_t} - p_{\tau_s} \right]. \tag{7}$$

This result extends to multiple tokens by summing up the per-token contributions. Here, $J_\theta = \partial z / \partial \theta$ is the Jacobian with $z$ being the logits, $p_{\tau_t} = \text{softmax}(z/\tau_t)$ is the training distribution, and $p_{\tau_s} = \text{softmax}(z/\tau_s)$ is the sampling distribution. By setting $\tau_s < \tau_t$ and with the same logits in both softmaxes, $p_{\tau_s}$ is a shaper distribution than $p_{\tau_t}$. Consequently, the vector $p_{\tau_t} - p_{\tau_s}$ is negative for the model's most preferred prediction, while it is typically positive on the others. Updating the model by one gradient step enhances the certainty of the model's existing preference.

## 4 EXPERIMENTS

### 4.1 EXPERIMENT SETUP

To evaluate the effectiveness of OSFT, we conduct a comparative analysis against GRPO, a popular on-policy RL algorithm, to benchmark its self-help capabilities.

**Datasets.** Our primary training question/prompt set is DeepSclaR (Luo et al., 2025). We evaluate all models on a suite of six challenging mathematical reasoning benchmarks: Math500 (Hendrycks et al., 2021), AMC (Li et al., 2024), Minerva math (Minerva) (Lewkowycz et al., 2022), Olympiad-Bench (Olympiad) (He et al., 2024), AIME24 (Li et al., 2024), and AIME25 (Li et al., 2024).

**Template and Evaluation.** To ensure a clean comparison, all experiments use the official Qwen2.5-Math system prompt without any additional user prompts, as detailed in Figure 10. Unless otherwise specified for ablation studies, the default evaluation temperature $\tau_{eval}$ is set to 1, which corresponds to the model's original output distribution. We evaluate performance using the pass@k metric. To ensure a stable and reliable measurement, we employ both pass@1 and pass@8 performance metrics, and the details are provided in Appendix A.2.

**Hyperparameters.** For OSFT, we use a decoupled temperature setting: The sampling temperature is $\tau_s = 0.6$ for the specialized Qwen math models and $\tau_s = 0.9$ for other models, while the training temperature is fixed at $\tau_t = 1$. For the GRPO baseline, we follow standard practice and use a coupled temperature setting of $\tau_s = \tau_t = 1$ (Sheng et al., 2025). An important difference highlighting OSFT's efficiency is the number of rollouts per prompt ($G$): Our default setting for OSFT is $G = 1$, whereas for GRPO it is $G = 8$.

All experiments are conducted using the VERL framework (Sheng et al., 2025). Comprehensive training and evaluation configurations are provided in Appendix A.

### 4.2 OSFT FOR LLM REASONING: ANALYSIS AND PERFORMANCE

#### 4.2.1 PROBABILITY ANALYSIS OF EXISTING PREFERENCE AND THE ROLE OF OSFT

We analyze the performance of the base Qwen2.5-Math-7B model, $\pi_\theta$, and the OSFT-trained model, $\pi_{\hat{\theta}}$, on a math problem in the Math500 dataset. As shown in Figure 2, the base model consistently

generates the incorrect path $[A, B]$. The OSFT model, in contrast, learns to generate the correct path $[\hat{A}, \hat{B}]$. The full question and responses of both models are put in Appendix D. We observe that $A$ and $\hat{A}$ (also $B$ and $\hat{B}$) are semantically similar.

The failure of the base model $\pi_\theta$ stems from its uncertainty in its own existing preference obtained from pretraining. First, while it generates path $A$, its own distribution actually shows a slight preference for the alternative prefix $\hat{A}$ ($\log \pi_\theta(\hat{A}|q) = -53$ vs. $-55$). The stochastic nature of sampling leads it to select the slightly less probable, and suboptimal, starting path. Once committed to prefix $A$, the model generates the incorrect suffix $B$ despite assigning a far higher probability to the correct suffix $\hat{B}$ ($\log \pi_\theta(\hat{B}|q, A) = -61$ vs. $-109$). The base model already possesses the latent knowledge that starting with $\hat{A}$ provides a much clearer path to the correct answer (as shown by the high probability of $\log \pi_\theta(\hat{B}|q, \hat{A}) = -49$), but it is unable to reliably follow this correct trajectory.

After training, the OSFT model $\pi_{\hat{\theta}}$ learns to overcome this uncertainty by systematically reinforcing the model's existing latent knowledge. First, it promotes a large preference for the superior prefix $\hat{A}$ over $A$, significantly widening the log-probability gap from 2 to 46 ($\log \pi_{\hat{\theta}}(\hat{A}|q) = -62$ vs. $-108$). This makes the correct start highly preferable. Second, OSFT largely increases the log-probability margin between the correct and incorrect suffixes. For the chosen path starting with $\hat{A}$, the log-probability gap between the correct suffix $\hat{B}$ and the incorrect one $B$ increases from 72 in the base model to 121 after OSFT. This dynamic, which widens the margin to avoid the flawed paths, shares similarity with contrastive alignment methods like DPO (Rafailov et al., 2023). OSFT thus succeeds not by teaching the model new mathematical facts, but by aligning its generative process with the superior reasoning paths that were already latent/existing within its own distribution obtained from pretraining.

### 4.2.2 CERTAINTY METRICS: PERPLEXITY PERSPECTIVE

To quantify the model's certainty, we compute the perplexity (PPL) on a per-benchmark basis. Let $\mathcal{D}_{\text{benchmark}} = \{(q_i, o_i)\}_{i=1}^N$ be the set of $N$ generated prompt-response pairs using $\pi_\theta$ for a single benchmark. PPL is defined as

$$\text{PPL}(\mathcal{D}_{\text{benchmark}}; \pi_\theta) = \exp\left(\frac{-\sum_{i=1}^N \sum_{t=1}^{|o_i|} \log \pi_\theta(o_{i,t} \mid q_i, o_{i,<t})}{\sum_{i=1}^N |o_i|}\right).$$

PPL is a metric measuring the averaged probability of a generated response. A lower PPL score on a given benchmark indicates higher model certainty on that specific task distribution.

In Figure 3, we display the PPL across all six test benchmarks for $N = 16$ data pairs per benchmark, where the responses are generated by models trained using several algorithms. Interestingly, GRPO and its variants also drive the PPL down, which suggests that GRPO might partially enhances the model's existing preference for improved reasoning. A detailed investigation is beyond the scope of this work and is left for future work.

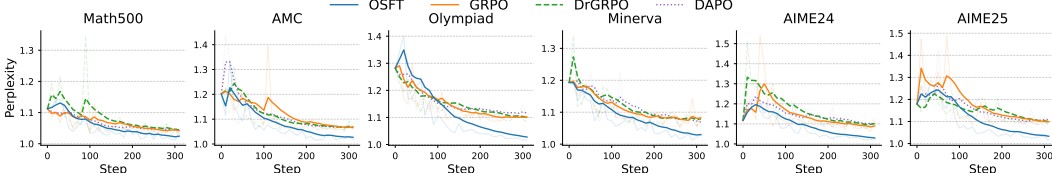

Figure 3: PPL of models trained using OSFT, GRPO, DAPO, and Dr. GRPO, where the base model is Qwen2.5-Math-7B.

### 4.2.3 QWEN-MATH SERIES PERFORMANCE

For downstream performance, we evaluate OSFT against the widely-used baseline GRPO (and its variants) on the specialized Qwen2.5-Math models at both 1.5B and 7B scales; see Figure 4.

The first observation lies in that in most cases, both OSFT and GRPO surpasses the base model's peak performance obtained by sweeping the evaluation temperature. This demonstrates that these

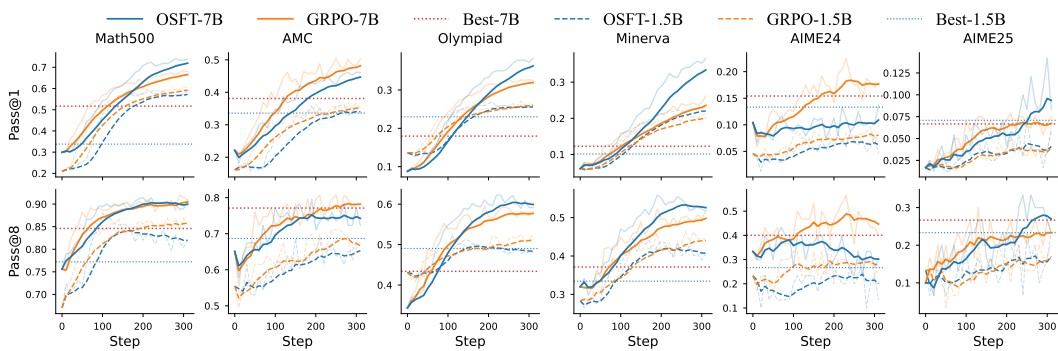

Figure 4: Performance of OSFT and GRPO on Qwen2.5-Math 1.5B (dashed lines) and 7B (solid lines) models on six math reasoning benchmarks. Dotted horizontal lines represent the best performance of the corresponding base models (before training) achieved by sweeping the evaluation temperature. It can be observed that OSFT is highly competitive with GRPO across different model scales on different math benchmarks.

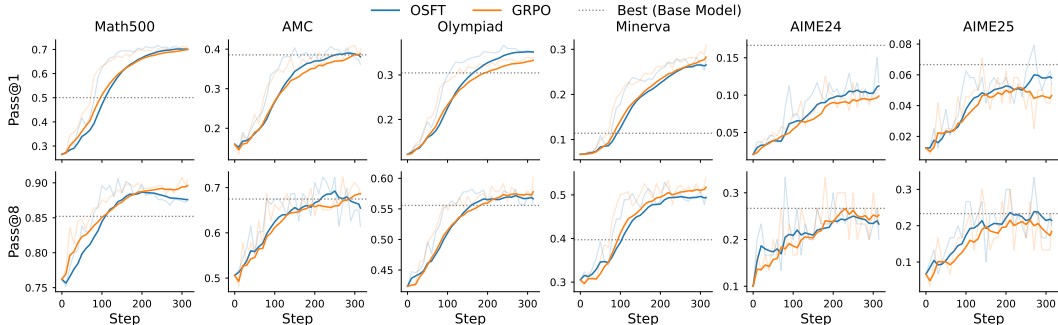

Figure 5: Performance of OSFT and GRPO on the Qwen2.5-7B base model across six math reasoning benchmarks. OSFT holds a comparable performance compared to GRPO.

online self-help algorithms provide essential and consistent performance gains over the base models. Additionally, OSFT exhibits a highly competitive mathematical reasoning performance with the strong baseline GRPO. For the 7B model, OSFT often matches or slightly surpasses GRPO's performance, particularly on benchmarks like Math500, Olympiad, and Minerva. This trend indicates that our simple, reward-free approach can achieve performance comparable to a more sophisticated RL algorithm. For a complete comparison against other RL baselines like DAPO and Dr. GRPO, please see Appendix B.1; they are excluded here to maintain visual clarity.

### 4.2.4 PERFORMANCE ON OTHER NON-MATH MODELS

To test OSFT's general ability, we apply it to the Qwen2.5-7B base model, which is not specialized for mathematical reasoning. The results are shown in Figure 5.

First, both OSFT and GRPO demonstrate the ability to improve the base model's best performance, with training curves rising well above the temperature-swept baselines (dotted lines) on MATH-500, Olympiad, and Minerva benchmarks. Overall, OSFT's performance curve is highly comparable to that of the strong baseline GRPO. Even in the challenging AIME benchmarks, OSFT's upward learning trend consistently mirrors that of GRPO. This validates that our simpler, reward-free method can be as effective as the more complex RL counterpart on a general-purpose base model.

We also provide experiments on the Llama model (Llama3.1-8B-Instruct). The result is shown in Appendix B.3. The conclusion is that OSFT can be comparable to GRPO on other model architectures like Llama.

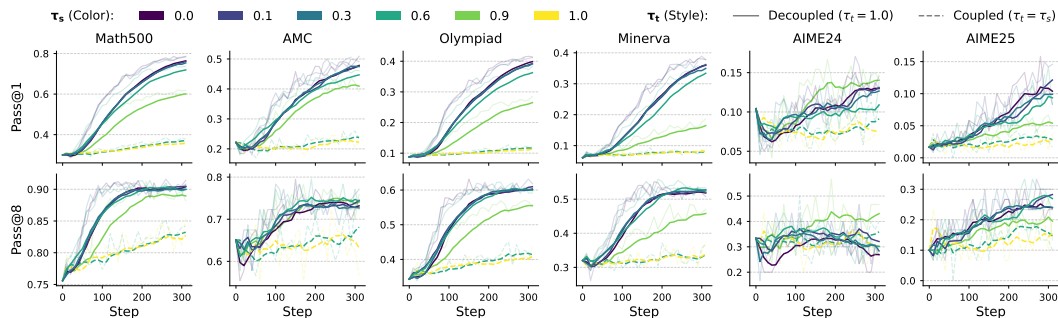

Figure 7: Ablation study on the decoupled temperature dynamics in OSFT. The figure illustrates the impact of sampling temperature ($\tau_s$, color) and the training temperature configuration ($\tau_t$, line style). The line style distinguishes the superior decoupled setting ($\tau_s < \tau_t = 1$) from the unstable coupled setting ($\tau_s = \tau_t$), providing strong empirical validation for our discussion in Section 3.3.

#### 4.2.5 OTHER TRAINING QUESTION SET

To evaluate the impact of the training data scope, we substitute the default dataset DeepScaleR with the Openthoughts (Guha et al., 2025) math-only (OpenthoughtsMath) dataset. The best performance achieved by each method within the first 300 training steps is shown in Figure 6. It can be seen that the new question set provides slightly worse performance for both OSFT and GRPO. This reveals that OSFT has a comparable level of generalization ability to different datasets relative to GRPO.

### 4.3 ABLATION STUDY

In this section, we conduct ablation studies on different hyperparameters, e.g., the different temperatures $\tau_s, \tau_t, \tau_{eval}$, and the number of rollouts $G$. All the ablation study experiments in this subsection are conducted by using Qwen2.5-Math-7B as the base model.

#### 4.3.1 DIFFERENT SAMPLING AND TRAINING TEMPERATURES FOR OSFT

The results shown in Figure 7 provide strong empirical validation for our discussion in Section 3.3. The coupled temperature setting ($\tau_t = \tau_s$, dashed lines) consistently fails to provide meaningful improvement. This confirms our discussion that when $\tau_t = \tau_s$, the learning signal degenerates into a random gradient noise update with no consistent direction. In the decoupled setting $\tau_s < \tau_t = 1$, we observe consistent improvement, while the value of $\tau_s$ is important for the final performance.

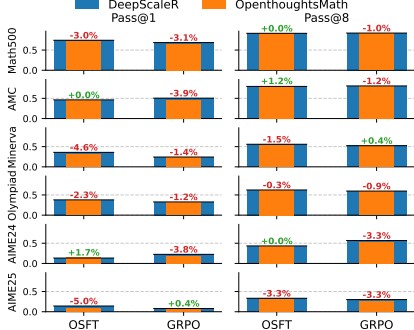

Figure 6: Performance impact of training data source. Peak scores (within 300 steps) are compared for models trained on DeepScaleR (blue baseline) versus Openthoughts math-only (orange). Percentages show the performance change from using OpenthoughtsMath.

#### 4.3.2 ABLATION ON THE NUMBER OF ROLLOUTS PER PROMPT

We investigate the effect of the number of self-generated samples per prompt, denoted by $G$ in Algorithm 1, on OSFT's performance. We compare our default data-efficient setting of $G = 1$ against a more data-intensive setting of $G = 4$.

The results, presented in Figure 8, reveals that $G = 4$ can indeed consistently improve the pass@1 performance of OSFT, while the pass@8 performance of these two settings remain nearly the same. We use $G = 1$ as our default setting in the comparison with GRPO ($G = 8$) to ensure that OSFT is much more time efficient.

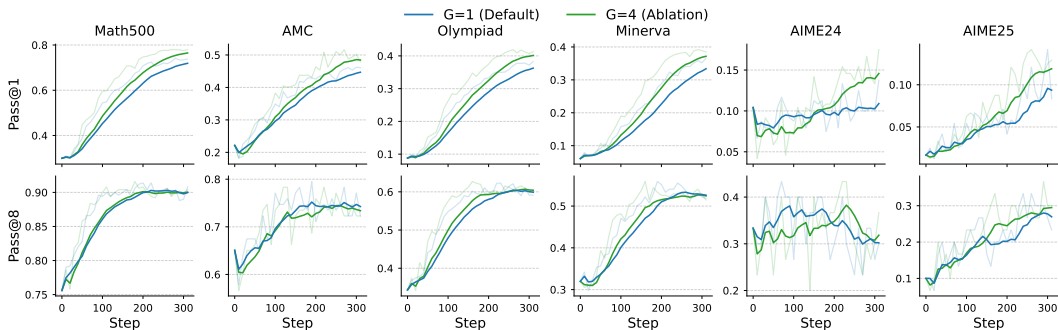

Figure 8: Ablation study on the number of self-generated samples ($G$) per prompt in OSFT.

### 4.3.3 ABLATION ON THE EVALUATION TEMPERATURE

We use the default $\tau_{eval} = 1$ in the main experiments to isolate the effect of artificially added certainty during inference, as $\tau_{eval} = 1$ corresponds to the model's original output distribution. We now investigate the influence of different evaluation temperatures ($\tau_{eval} \in \{0.1, 0.3, 0.6, 1\}$) on the step-300 checkpoints obtained by both OSFT and GRPO.

The results are shown in Figure 9. The pass@1 performance still peaks at lower temperatures, while pass@8 tends to benefit from higher temperatures (except for AIME24). The shift in pass@8's preference compared to Figure 1a is likely due to the trained models becoming more certain than the base model. Interestingly, both OSFT and GRPO exhibit remarkably similar variation trends across benchmarks. Despite their fundamentally different training schemes, both methods converge to final models with highly similar output distributions in terms of solving downstream tasks. This further illustrates that OSFT is a robust and comparable alternative to GRPO.

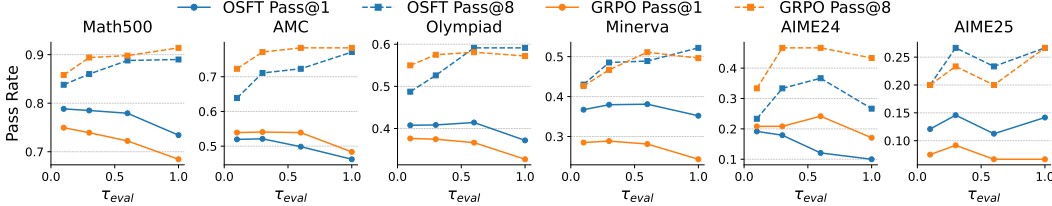

Figure 9: Ablation study of evaluation temperature $\tau_{eval}$ for both OSFT and GRPO. Both OSFT and GRPO exhibit remarkably similar variation trends across benchmarks

We conclude this section by noting that OSFT achieves downstream performance comparable to GRPO across different models and training question sets. The ablation study further confirms the efficiency of OSFT. In summary, we believe that OSFT presents a promising and competitive alternative for LLM reasoning.

## 5 CONCLUSION

In this work, we introduced the OSFT training strategy, a self-help algorithm for LLM reasoning. OSFT is highly efficient, as it is reward-free and uses only one rollout by default. We discussed the importance of temperature decoupling in OSFT. Our experiment results demonstrated that OSFT can achieve downstream performance comparable to the strong RL baseline GRPO. The ablation studies confirmed the importance of temperature decoupling and showed the efficiency and robustness of OSFT.

Our discussion and analysis experiments illustrated that OSFT enhances the base model's existing preferences learned during pretraining, leading to improved reasoning ability. A similar trend of increased certainty was also observed in GRPO and its variants. We leave deeper investigation into the relationship between OSFT and GRPO in reasoning ability improvement as future work.

## REPRODUCIBILITY STATEMENT

To ensure the reproducibility of our results, we provide a comprehensive suite of materials. The source code for our Online SFT (OSFT) algorithm and main experiments is available in the supplementary materials as an anonymized package. Our method is formally described in Section 3, with the core algorithm detailed in Algorithm 1. A thorough description of the experimental setup, including datasets, models, and evaluation protocols, is provided in Section 4.1. All hyperparameters and detailed configurations for both our method and the baselines are documented in Appendix A. Additional analyses, including ablation studies and results on different model architectures, can be found in Section 4.3 and Appendix B. Furthermore, to guide a successful replication effort, we discuss crucial factors that can influence performance, such as verifier logic and system-level non-determinism, in Appendix F.

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

CONTENTS

## USE OF LARGE LANGUAGE MODELS

Throughout the writing process for this paper, we used Google's Gemini 2.5 Pro and OpenAI's GPT-4o to polish writing. The AI model's contributions were strictly limited to rephrasing and polishing the text for grammar, clarity, and conciseness. All scientific ideas, experimental designs, and analyses were conceived and executed by the human authors. The authors have reviewed and take full responsibility for the entire content of this paper.

## A   DETAILED EXPERIMENT SETUP

### A.1   TRAINING CONFIGURATION

**General Parameters.** All models are trained on a cluster of 8 NVIDIA A800 GPUs. For the 7B and 8B models, we use a constant learning rate of 1e-7, while for the 1.5B model, we use 3e-7. All training runs include a 10-step warmup period. We follow recent works (Hu et al., 2025b; Yu et al., 2025; Liu et al., 2025), and to ensure a direct comparison of the core learning algorithms, the KL divergence regularizer is disabled for all experiments.

**Batching and Gradient Updates.** All experiments share a common batching structure. The global batch size is set to 128 prompts per training step. These prompts are processed over two gradient updates, with a mini-batch size of 64 prompts per update. The micro-batch size is fixed at 32 sequences per GPU for each forward/backward pass. The total number of sequences per update and the resulting number of gradient accumulation steps depend on the number of rollouts ($G$), which is method-specific.

**Method-Specific Configurations.**

**OSFT.** Our method is configured for efficiency, using a single rollout per prompt ($G = 1$). Consequently, each gradient update processes $64 \times 1 = 64$ sequences. This batch fits within a single forward pass across all GPUs, so no gradient accumulation is needed. OSFT employs a decoupled temperature scheme: The training temperature is fixed at $\tau_t = 1.0$, while the sampling temperature $\tau_s$ is set to 0.6 for specialized math models and 0.9 for general-purpose base models.

**GRPO.** The GRPO baseline uses the standard 8 rollouts per prompt ($G = 8$) and a coupled temperature of $\tau_s = \tau_t = 1.0$. Each gradient update processes a total of $64 \times 8 = 512$ sequences. Given the micro-batch size of 32 and 8 GPUs, this requires 2 gradient accumulation steps per update ($512/(32 \times 8) = 2$).

**GRPO Variants.** For DAPO and Dr. GRPO, the batching logic and rollout number ($G = 8$) are identical to the GRPO baseline. We follow the recommended hyperparameter settings from the VERL framework. For **DAPO**, we set the clipping ratios to 'clip-ratio-c=10', 'clip-ratio-low=0.2', and 'clip-ratio-high=0.28'. For **Dr. GRPO**, we disable standard deviation normalization and use the 'seq-mean-token-sum-norm' method for loss aggregation.

### A.2   EVALUATION CONFIGURATION

In our experiments, we report both pass@1 and pass@8 performance metrics using $n = 8$ generated samples per prompt. For each prompt, pass@1 is computed as $c/n$, where $c$ is the number of correct responses. The final pass@1 score is then averaged over all prompts in the benchmark. Since $n = 8$, pass@8 for a prompt is 1 if at least one response out of the 8 responses is correct, and 0 otherwise. The final pass@8 score is also averaged over all prompts in the benchmark.

For our main experiments, we set the evaluation temperature to $\tau_{eval} = 1$, except when conducting specific temperature sweep analyses. This standard setting is chosen to evaluate the model's performance based on its unaltered probability distribution. We note that using lower evaluation temperatures may boost pass@1 accuracy; for instance, the released code of prior work (Shao et al., 2025) has used value as low as 0. By using $\tau_{eval} = 1$ as our default, we aim to provide a direct assessment of the model's original capabilities as learned during training, without post-hoc optimization of decoding parameters. We also provide ablation study for $\tau_{eval}$ in Section 4.3.3.

### A.3 VERIFIER

The selection of a verifier is an important factor for the final score. Various verifiers may use different methods for parsing answers (parser) and assessing correctness (grader). Considering the trade-off between response time and accuracy, we evaluated multiple verifiers and chose the math verifier presented in Cui et al. (2025).

### A.4 VISUALIZATION

To improve readability and highlight underlying trends in our training curves, most performance plots are smoothed using an Exponential Moving Average (EMA) with a span of 10. The raw, unsmoothed data points are shown transparently in the background to ensure a full disclosure of the original performance dynamics.

### A.5 CHAT TEMPLATE

To ensure proper model interaction and reproducibility, we adhere to the clean official chat template for Qwen math model, while using each model family's own special tokens to structure the conversational roles for the system, user, and assistant. The specific templates used for the Qwen and Llama families are shown in Figure 10.

---

**Qwen family**

```
<|im_start|>system
{System}<|im_end|>
<|im_start|>user
{User}<|im_end|>
<|im_start|>assistant
```

---

**Llama family**

```
<|begin_of_text|><|start_header_id|>system<|end_header_id|>

{System}<|eot_id|><|start_header_id|>user<|end_header_id|>

{User}<|eot_id|><|start_header_id|>assistant<|end_header_id|>
```

---

Figure 10: Chat template, including special tokens, for the Qwen-2.5 and Llama-3.1 series. The system prompt is consistent across all models: "Please reason step by step, and put your final answer within \boxed{}". The user prompt is the math problem from the dataset, without any additional artificially created user prompts.

# B ADDITIONAL EXPERIMENT RESULTS

This section provides supplementary results and analyses that support the claims made in the main paper.

## B.1 PERFORMANCE CURVES ON OTHER GRPO VARIANTS

To provide a more comprehensive benchmark, we compare OSFT not only against the standard GRPO but also against its popular variants such as DAPO and Dr. GRPO. The performance curves for all four methods are shown in Figure 11.

A key observation from these results is that the performance of GRPO and its variants (DAPO, Dr. GRPO) is similar under our experimental conditions. The learning trajectories for these three RL-based methods are nearly indistinguishable across the benchmarks.

Our simple, reward-free OSFT method demonstrates a comparable performance. Its learning curves consistently track alongside the cluster of GRPO methods, indicating that OSFT is a compelling and efficient alternative for LLM reasoning.

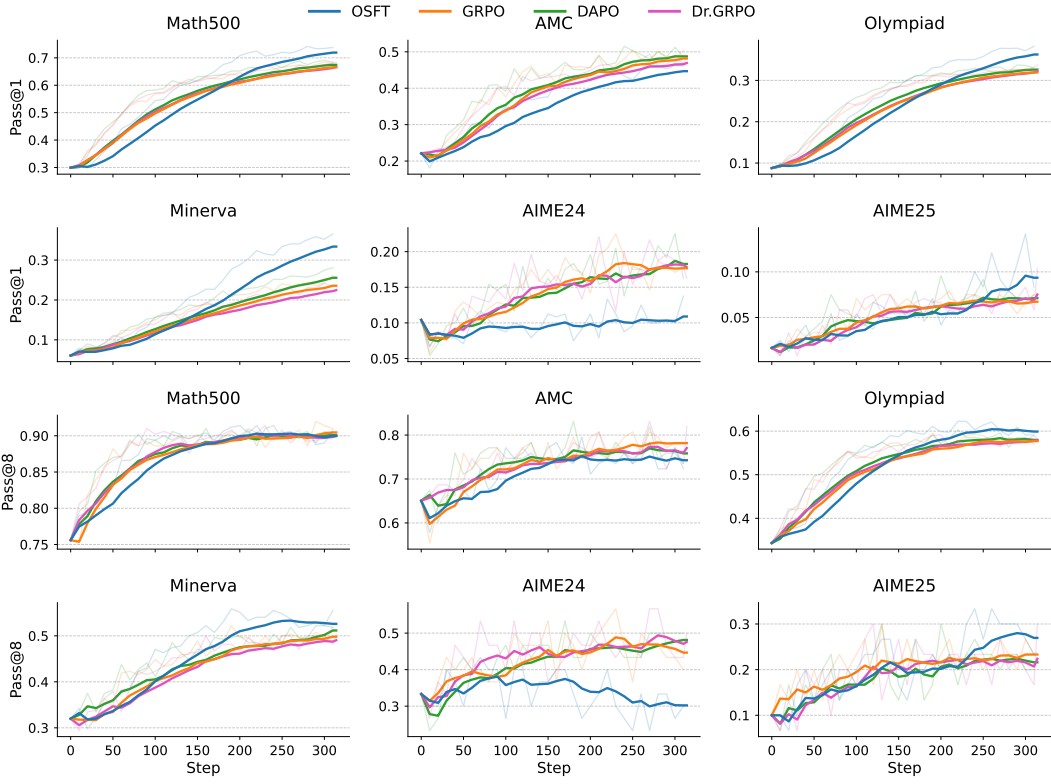

Figure 11: Performance comparison of OSFT against GRPO and its variants (DAPO, Dr. GRPO) on the Qwen2.5-Math-7B model. The plots show pass@1 (top 2 rows) and pass@8 (bottom 2 rows) performance. The performance of GRPO, DAPO, and Dr. GRPO is similar in this setting. OSFT achieves results comparable to all three RL methods.

## B.2 ABLATION ON SAMPLING TEMPERATURE FOR THE BASE MODEL

As discussed in the main text, the sampling temperature ($\tau_s$) is an important factor for OSFT's final performance. We use $\tau_s = 0.6$ for Qwen Math models, while $\tau_s = 0.9$ for other general-purpose base models. We discuss here why we need a higher $\tau_s$ for general-purpose base models. For the Qwen2.5-7B, Figure 12 provides an empirical validation of our selection of $\tau_s = 0.9$, showing that lower values lead to performance degradation in later training stages on the benchmarks.

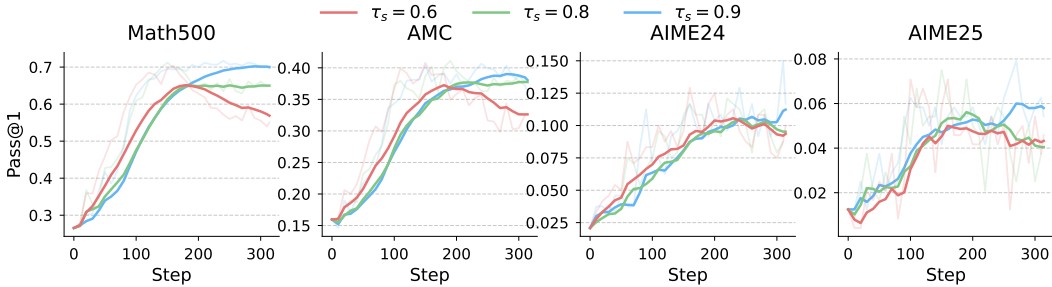

Figure 12: Empirical validation for the choice of a higher sampling temperature ($\tau_s$) on the general-purpose non-math Qwen2.5-7B model. Pass@1 performance is shown for four representative benchmarks. The setting used in the main paper, $\tau_s = 0.9$, maintains stable improvement. In contrast, lower values lead to performance degradation in later training stages.

### B.3 PERFORMANCE ON LLAMA3.1-8B-INSTRUCT

To assess the generalizability of our method beyond the Qwen architecture, we applied OSFT and GRPO to the Llama3.1-8B-Instruct model. We chose the instruction-tuned variant as the base model exhibited difficulty in following the required problem-solving format.

The quantitative results are shown in Table 1, the peak performance difference between OSFT and GRPO are comparable, suggesting that both methods perform similarly on model architectures other than Qwen family.

The experiment results also suggest that while OSFT and GRPO can improve performance on different model architectures like Llama, their effectiveness is highly related to the foundational capabilities of the base model. The modest gains on Llama3.1-8B-Instruct, compared to the stronger improvements on Qwen models, highlight that base model is also an important factor for achieving substantial reasoning improvement through RL techniques and OSFT. This aligns with recent observations in other works; see, e.g., Liu et al. (2025).

Table 1: Quantitative comparison of peak performance for training Llama-3.1-8B-Instruct. The table shows the highest scores achieved by OSFT and GRPO. The performance difference between the two methods are comparable across all benchmarks.

| Benchmark | Metric | Peak Score | | $\Delta$ (OSFT - GRPO) |
| --- | --- | --- | --- | --- |
| | | **GRPO** | **OSFT** | |
| Math500 | pass@1 | 0.533 | 0.504 | -0.029 |
| | pass@8 | 0.804 | 0.782 | -0.022 |
| AMC | pass@1 | 0.244 | 0.255 | +0.011 |
| | pass@8 | 0.578 | 0.542 | -0.036 |
| Olympiad | pass@1 | 0.191 | 0.169 | -0.021 |
| | pass@8 | 0.413 | 0.404 | -0.009 |
| Minerva | pass@1 | 0.267 | 0.272 | +0.005 |
| | pass@8 | 0.518 | 0.485 | -0.033 |
| AIME24 | pass@1 | 0.092 | 0.087 | -0.004 |
| | pass@8 | 0.333 | 0.333 | +0.000 |
| AIME25 | pass@1 | 0.013 | 0.013 | +0.000 |
| | pass@8 | 0.100 | 0.100 | +0.000 |

## C   ONLINE SFT WORKFLOW

This section provides a visual depiction of OSFT training strategy, as illustrated in Figure 13. The process is an iterative cycle consisting of two main phases: a self-generation step where the model creates its own training data, followed by a SFT step where the model learns from that self-generated data.

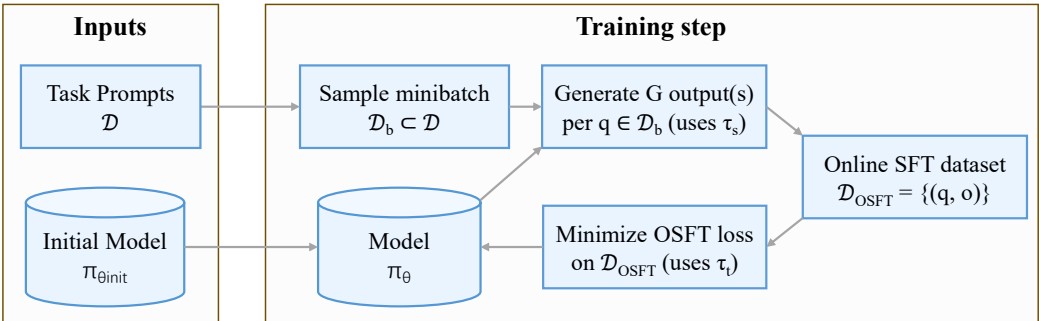

Figure 13: The iterative workflow of OSFT. The model alternates between generating its own training data and finetuning on it.

## D  FULL QUESTION AND RESPONSES FROM BASE AND OSFT MODELS

This section provides the full text for the case study visualized in Figure 2. Figure 14 details one incorrect response chosen from all 8 wrong tries from the base model, while Figure 15 shows the corresponding correct response generated after performing OSFT.

---

**Example from Math500**

**Problem**: The expression $2 \cdot 3 \cdot 4 \cdot 5 + 1$ is equal to 121, since multiplication is carried out before addition. However, we can obtain values other than 121 for this expression if we are allowed to change it by inserting parentheses. For example, we can obtain 144 by writing

$$(2 \cdot (3 \cdot 4)) \cdot (5 + 1) = 144.$$

In total, how many values can be obtained from the expression $2 \cdot 3 \cdot 4 \cdot 5 + 1$ by inserting parentheses? (Note that rearranging terms is not allowed, only inserting parentheses).

---

**Response (Base Model)**

To determine how many distinct values can be obtained from the expression $2 \cdot 3 \cdot 4 \cdot 5 + 1$ by inserting parentheses, we need to consider all possible ways of inserting parentheses into the expression.
First, let's evaluate the expression without any parentheses:

$$2 \cdot 3 \cdot 4 \cdot 5 + 1 = 121$$

Now, we need to consider all possible ways of inserting parentheses.

There are five possible ways to insert one set of parentheses: 1. $((2 \cdot 3) \cdot 4) \cdot 5 + 1$ 2. $(2 \cdot (3 \cdot 4)) \cdot 5 + 1$ 3. $(2 \cdot 3) \cdot (4 \cdot 5) + 1$ 4. $2 \cdot ((3 \cdot 4) \cdot 5) + 1$ 5. $2 \cdot (3 \cdot (4 \cdot 5)) + 1$
However, the first and fifth expressions will be the same, as will the second and fourth. This leaves us with three unique ways of inserting parentheses, in addition to the original expression without parentheses.
So we need to evaluate each of these expressions: 1. $((2 \cdot 3) \cdot 4) \cdot 5 + 1 = (6 \cdot 4) \cdot 5 + 1 = 24 \cdot 5 + 1 = 121$ 2. $(2 \cdot (3 \cdot 4)) \cdot 5 + 1 = (2 \cdot 12) \cdot 5 + 1 = 24 \cdot 5 + 1 = 121$ 3. $(2 \cdot 3) \cdot (4 \cdot 5) + 1 = 6 \cdot 20 + 1 = 120 + 1 = 121$ 4. $2 \cdot ((3 \cdot 4) \cdot 5) + 1 = 2 \cdot (12 \cdot 5) + 1 = 2 \cdot 60 + 1 = 120 + 1 = 121$ 5. $2 \cdot (3 \cdot (4 \cdot 5)) + 1 = 2 \cdot (3 \cdot 20) + 1 = 2 \cdot 60 + 1 = 120 + 1 = 121$
From this, we see that all possible ways of inserting parentheses result in the value 121. Therefore, the only distinct value we can obtain is 121.
Hence, the number of distinct values we can obtain from the expression $2 \cdot 3 \cdot 4 \cdot 5 + 1$ by inserting parentheses is:

$$\boxed{1}$$

---

Figure 14: Full question and the incorrect response generated by the base model, corresponding to the analysis in Figure 2. The soft blue and light gray backgrounds distinguish the initial reasoning prefix ($A$) from the subsequent flawed suffix ($B$). This segmentation is because $B$ contains the core reasoning and calculation leading to the answer, while $A$ represents the initial problem setup.

---

**Example from Math500**

**Problem**: The expression $2 \cdot 3 \cdot 4 \cdot 5 + 1$ is equal to 121, since multiplication is carried out before addition. However, we can obtain values other than 121 for this expression if we are allowed to change it by inserting parentheses. For example, we can obtain 144 by writing

$$(2 \cdot (3 \cdot 4)) \cdot (5 + 1) = 144.$$

In total, how many values can be obtained from the expression $2 \cdot 3 \cdot 4 \cdot 5 + 1$ by inserting parentheses? (Note that rearranging terms is not allowed, only inserting parentheses).

---

**Response (OSFT Model)**

To determine how many different values can be obtained from the expression $2 \cdot 3 \cdot 4 \cdot 5 + 1$ by inserting parentheses, we need to consider all possible ways to group the terms. Let's break this down step by step.

First, let's list all the possible ways to insert parentheses in the expression $2 \cdot 3 \cdot 4 \cdot 5 + 1$:

1. $(2 \cdot 3 \cdot 4 \cdot 5) + 1$ 2. $2 \cdot (3 \cdot 4 \cdot 5 + 1)$ 3. $2 \cdot 3 \cdot (4 \cdot 5 + 1)$ 4. $2 \cdot (3 \cdot (4 \cdot 5 + 1))$ 5. $(2 \cdot 3 \cdot 4) \cdot (5 + 1)$ 6. $(2 \cdot 3) \cdot (4 \cdot (5 + 1))$ 7. $2 \cdot ((3 \cdot 4) \cdot (5 + 1))$ 8. $((2 \cdot 3) \cdot 4) \cdot (5 + 1)$ 9. $(2 \cdot (3 \cdot 4)) \cdot (5 + 1)$
Now, let's evaluate each of these expressions:
1. $(2 \cdot 3 \cdot 4 \cdot 5) + 1 = 120 + 1 = 121$ 2. $2 \cdot (3 \cdot 4 \cdot 5 + 1) = 2 \cdot (60 + 1) = 2 \cdot 61 = 122$ 3. $2 \cdot 3 \cdot (4 \cdot 5 + 1) = 6 \cdot (20 + 1) = 6 \cdot 21 = 126$ 4. $2 \cdot (3 \cdot (4 \cdot 5 + 1)) = 2 \cdot (3 \cdot 21) = 2 \cdot 63 = 126$ 5. $(2 \cdot 3 \cdot 4) \cdot (5 + 1) = 24 \cdot 6 = 144$ 6. $(2 \cdot 3) \cdot (4 \cdot (5 + 1)) = 6 \cdot (4 \cdot 6) = 6 \cdot 24 = 144$ 7. $2 \cdot ((3 \cdot 4) \cdot (5 + 1)) = 2 \cdot (12 \cdot 6) = 2 \cdot 72 = 144$ 8. $((2 \cdot 3) \cdot 4) \cdot (5 + 1) = (6 \cdot 4) \cdot 6 = 24 \cdot 6 = 144$ 9. $(2 \cdot (3 \cdot 4)) \cdot (5 + 1) = (2 \cdot 12) \cdot 6 = 24 \cdot 6 = 144$
From the evaluations, we see that the possible values are 121, 122, 126, and 144. Therefore, there are 4 distinct values that can be obtained.
The final answer is $\boxed{4}$.

---

Figure 15: Full question and the correct response generated by the OSFT model, corresponding to the analysis in Figure 2. The soft blue and light gray backgrounds distinguish the initial reasoning prefix ($\hat{A}$) from the subsequent correct suffix ($\hat{B}$). This segmentation is because $\hat{B}$ contains the core reasoning and calculation leading to the answer, while $\hat{A}$ represents the initial problem setup.

# E  MISSING MATHEMATICAL DERIVATIONS

## E.1  SCORE-FUNCTION IDENTITY

The following derivation is the standard score-function identity, which shows that one step update of OSFT with $\tau_s = \tau_t = \tau$ is equivalent to a random gradient noise update.

$$
\begin{aligned}
\mathbb{E}_{o \sim \pi_\theta(\cdot | q)} \left[ \nabla_\theta \log \pi_\theta(o | q) \right] &= \sum_o \pi_\theta(o | q) \cdot \frac{\nabla_\theta \pi_\theta(o | q)}{\pi_\theta(o | q)} \\
&= \sum_o \nabla_\theta \pi_\theta(o | q) \\
&= \nabla_\theta \sum_o \pi_\theta(o | q) \\
&= \nabla_\theta 1 \\
&= 0.
\end{aligned}
\tag{8}
$$

## E.2  GRADIENT CALCULATION

We now derive the gradient displayed in (7). By chain rule, we have

$$
\frac{\partial \mathcal{L}}{\partial \theta} = \frac{z}{\partial \theta} \frac{\partial \mathcal{L}}{\partial z} = J_\theta \frac{\partial \mathcal{L}}{\partial z},
$$

where we have omitted $\theta$ in $z$ for simplicity. By the definition of softmax, for a specific class/token $o$ in the vocabulary we have

$$
p_{\tau_t}(o) = \frac{e^{z_o/\tau_t}}{\sum_j e^{z_j/\tau_t}}, \quad p_{\tau_s}(o) = \frac{e^{z_o/\tau_s}}{\sum_j e^{z_j/\tau_s}}.
$$

It follows that

$$
\log(p_{\tau_t}(o)) = \frac{z_o}{\tau_t} - \log \left( \sum_j e^{z_j/\tau_t} \right).
$$

Taking derivative over the class $k$'s logit $z_k$ provides

$$
\frac{\partial \log(p_{\tau_t}(o))}{\partial z_k} = \frac{1}{\tau_t} \left( 1_{\{k=o\}} - p_{\tau_t}(k) \right).
$$

Let us use $e_o$ to denote the unit vector with 1 at its $o$-th position. Then, the gradient over the whole logits $z$ is given by

$$
\frac{\partial \log(p_{\tau_t}(o))}{\partial z} = \frac{1}{\tau_t} \left( e_o - p_{\tau_t} \right).
$$

For the loss we have $\mathcal{L} = \mathbb{E}_{o \sim p_{\tau_s}}[- \log(p_{\tau_t}(o))]$. Therefore, we can further compute

$$
\nabla_z \mathcal{L} = \mathbb{E}_{o \sim p_{\tau_s}} \left[ \frac{1}{\tau_t}(p_{\tau_t} - e_o) \right] = \frac{1}{\tau_t}(p_{\tau_t} - p_{\tau_s}).
$$

This completes the derivation.

# F   Factors Influencing Evaluation Performance

The final reported performance metrics for all methods can be sensitive to several factors. These can be categorized into the evaluation logic, the generation configuration, and sources of non-determinism in the underlying system. We list several representative ones below.

1. **Verifier Logic:** The choice of verifier is an important factor the final score. Different verifiers may employ distinct methods for parsing answers (parser) and judging correctness (grader).

2. **Sampling Parameters:** The decoding strategy, governed by parameters such as temperature, top-p, top-k, and repetition penalty, shapes the output. While a deterministic strategy (e.g., greedy decoding, by simply setting temperature to 0) will produce a consistent output, whereas stochastic sampling introduces diversity/exploration.

3. **Random Seed:** The seed is important for reproducibility as it controls the stochastic sampling process. A different seed will result in a different sequence of sampled tokens, leading to a different generated output and thus a different final score.

4. **Tensor Parallelism:** The use of tensor parallelism introduces non-determinism, even with a fixed seed. This is a known consequence of floating-point arithmetic, where summation across distributed devices via communication collectives (e.g., `All-Reduce`) is not associative. Such differences in calculated logits can be sufficient to alter the final token selection, causing inconsistency in generated sequences between runs with and without tensor parallelism.

5. **GPU Architecture:** Different GPU hardware (e.g., NVIDIA A100 vs. H100) or underlying library versions (e.g., cuDNN) may implement fundamental operations with slight algorithmic variations. This can lead to small numerical discrepancies that propagate through the model, yielding different results.

6. **Order of Data:** The order of prompts within a batch can alter calculation results, especially in dynamic batching engines used in vLLM. This can lead to different outputs. Hence, we fix the data order for the test benchmark datasets to ensure reproducible evaluations.

