# OpenReview forum: "Online SFT for LLM Reasoning"
_ICLR.cc/2026/Conference — ICLR 2026 Conference Withdrawn Submission_

### Official Review · Reviewer_DLv4 · 2025-10-17

**Soundness:** 3
**Presentation:** 3
**Contribution:** 2
**Rating:** 4
**Confidence:** 4

**Summary:**

This paper proposes OSFT, a simple, self-help online supervised finetuning method for LLM reasoning. Specific, OSFT performs generation on training dataset and leverage the self-generated data to finetune the LLM. Emperical results on different mathematical benchmarks and diverse math-specific and general LLMs demonstrate the effectiveness of OSFT. Authors also conduct ablation studies to understand the mechanism of OSFT more deeply.

**Strengths:**

* A relatively simple and straightforward training method for LLM reasoning is proposed, which can achieve results comparable to RLVR methods such as GRPO while reducing training costs.
* Relatively extensive emperical experiments and detailed analysis.
* Good writing and structure, easy to read and follow.

**Weaknesses:**

* The core idea of the article is not a novel concept. For example, some existing verify-free works [1] all utilize the prior predictions of LLMs themselves on the training problem sets.
* Most of the conclusions in this paper are drawn from experiments on mathematical reasoning benchmarks, and their generalizability to reasoning tasks in other fields is questionable.
* In my opinion, a core disadvantage of OSFT is that it cannot bring about changes in the reasoning patterns of LLMs. That is, base LLMs cannot benefit from OSFT and learn to reason using long-CoT, which is the core contribution of works such as deepseek-r1.

[1] RLPR: Extrapolating RLVR to General Domains without Verifiers.

**Questions:**

* How is the generalization ability of OSFT for different reasoning tasks?
* Consider more extreme cases: if there is no correct data at all in the self-generated data, will OSFT fail to work? However, algorithms like GRPO can obtain correct responses by increasing the group size.

---

### Official Review · Reviewer_EhYb · 2025-10-27

**Soundness:** 2
**Presentation:** 2
**Contribution:** 2
**Rating:** 2
**Confidence:** 4

**Summary:**

This work proposes a self-learning approach, OSFT, which fine-tunes large language models (LLMs) using self-generated data. Compared with baselines such as GRPO, the proposed method demonstrates higher sample efficiency and better enhances the model’s reasoning ability. An interesting finding is that the temperature settings for data generation and fine-tuning should be decoupled. Specifically, the generation temperature should be lower than that used during fine-tuning to ensure stable optimisation. This insight is further supported by both theoretical proof and ablation studies.

**Strengths:**

* The proposed method is compared against GRPO variants, e.g. GRPO, DrGRPO, and DAPO, and evaluated on six different mathematical reasoning datasets.
* Comprehensive ablation studies are conducted to examine the impact of temperature settings during data generation, fine-tuning, and evaluation.

**Weaknesses:**

* The authors emphasise that their proposed method is reward-free, meaning that the generated results are not verified and may contain incorrect information. According to Algorithm 1, there is no filtering or validation step to ensure the correctness of the self-generated data. This raises concerns about potential performance degradation, particularly when training on challenging datasets where the model may rarely produce correct responses. Moreover, such an optimisation process could amplify existing model biases. These issues should be examined through additional experiments.
* Although experiments are also conducted on Llama-3.1-8B, the results presented in Appendix B.3 are incomplete. Evaluating the proposed method on a broader range of LLMs, such as Phi-2 and Gemma, would further strengthen the work. Given that a model’s intrinsic knowledge substantially affects its ability to learn from self-generated data, assessing models with varying reasoning abilities is essential to understand the generalisability and limitations of the approach.
* The study lacks certain baseline comparisons. As noted by the authors at the end of Section 4.2.1, DPO represents a relevant alternative and should be included to provide a more comprehensive evaluation. Including such baselines would enhance the credibility of the results, as the current comparisons focus solely on GRPO variants.

**Questions:**

* In Figure 2, it is unclear why the base model selects the path with a lower probability. Could this be a result of non-deterministic sampling during the generation process? The figure would benefit from additional explanation and clarification of this behaviour.
* In Section 4.2.2, it is mentioned that lower perplexity corresponds to better reasoning ability. However, perplexity primarily measures a model’s capacity to predict the next token in language modelling, rather than its logical validity or reasoning correctness. Therefore, further justification is needed to support the use of perplexity as an indicator of reasoning performance.
* It would be valuable to discuss connections with related work, such as Chain-of-Thought (CoT) decoding [1], which similarly observed that models can produce better responses even when those responses have lower probability. Drawing this comparison could help clarify the motivation and contextualise the proposed approach within existing literature.

[1] Chain-of-Thought Reasoning Without Prompting (Wang et al., NeurIPS 2024)

---

### Official Review · Reviewer_khgs · 2025-11-02

**Soundness:** 3
**Presentation:** 2
**Contribution:** 2
**Rating:** 4
**Confidence:** 3

**Summary:**

* **Approach**: This paper introduces an self-supervised alternative for RL-based approaches. The proposed approach OSFT which uses a careful observation about the difference in temperature used for self-supervised training data generation and model finetuning training temperature to get the method to work well. OSFT is noted for its efficiency (e.g., single roll out).
* **Empirical Succeses**: OSFT performs as well or better than GRPO for in many of the evaluations done by the authors, including on : Math500, AMC, Minerva math,. OlympiadBench, AIME24, AIME25.

**Strengths:**

* **Empirical Analysis**: The paper has a nice section of empirical analysis of the methods. This yields a fairly significant result in terms of the empirical performance of OSFT. For instance, Figure 3-7 seem to give a fairly clear understanding of the method's performance characteristics.
* **Efficiency of Method**: The paper's contribution of a highly accurate method for training reasoning models is notable because of its efficiency. This is an important contribution, it also can motivate future work in this important area.

**Weaknesses:**

* **Presentation concerns**: It's not clear to me why the method is named Online SFT. I find the use of "Online" quite confusing. Is it really just a single pass over the training dataset? Sorry if I have missed this in the paper. Online methods typically make a prediction, get feedback, update parameters & cannot return to that data point, e.g. single epoch over data. The algorithm box here has a loop over a number of steps and sampling of a mini batch. Also typical online methods get supervised feedback I thought. Here it seems more like standard self-supervised approach? For me, I found this very confusing and unsettling reading the paper.
* **Contribution Depth**: This is a nice, simple method, and the temperature observation is critical. But the technical differences between this method and standard self-distillation kinds of approaches seems to be quite small? I would have expected either much more depth of analysis of the method (theoretically, empirically) or a method that deviates more from existing approaches.

Minor:

**Questions:**

1. Please elaborate on differences between this approach and self-supervised approaches?
2. Have I missed something about why the word Online is used to describe the method?
3. Is the time difference between G=1 and G=4 a 4x time difference?

---

### Official Review · Reviewer_n1oD · 2025-11-02

**Soundness:** 1
**Presentation:** 3
**Contribution:** 2
**Rating:** 2
**Confidence:** 4

**Summary:**

The paper introduces Online Supervised Finetuning (OSFT), a "reward-free" method to improve LLM reasoning by finetuning a model on its own generated data (lines 189-192). The main idea is to use a simple SFT loss on data sampled at a low temperature ($\tau_s$) to train the model at a higher temperature ($\tau_t$).

**Strengths:**

The presentation is clear and easy to understand, with fairly extensive evidences (plots, math derivations). Details related to reproduction are good.

The experiment is fairly extensive, with 2 training datasets (deepscaler, openthoughts), 6 test datasets (math500, amc, minerva, olympiad, aime24, aime25), and 4 models (qwen2.5-math 1.5B and 7B, qwen2.5-7B, and llama3.1-8B-instruct) in total. 3 baseline algorithms are grpo, dr-grpo and dapo).

**Weaknesses:**

My main concern is the technical soundness and also some missing but important details.

My understanding is that you simply finetune on generated samples regardless of their correctness, thus "reward-free" and "just one rollout by default" (line 12). My intution is that the resulting model's behavior at high temperature will just mirror the untrained models' behavior at low temperature. In particular, the behavior should be like figure 1a. Truely, the model will become more confident (at high temperature vs. the untrained model), but it can also be more confident about the wrong answer too.

The clarifying evidences are weak.
1. The analysis in Figure 2 and Section 4.2.1 simply restates what a successful algorithm would achieve (widen probability gaps) rather than providing any new insight into why your algorithm leads to correct answers.
 2. The perplexity experiment (Section 4.2.2, lines 290 - 307). It is rather obvious that perplexity should go down for all listed algorithms on the generated data from untrained (?) model, because they are all fitting on samples from that distribution (in your case, both positive and negative, so even more effective).

The experimental results are quite mixed, the proposed method is significantly worse on AIME24 (figure 4 and 5), making the efficiency claim weak (you only need 1 rollout per prompt vs.8 for GRPO as you don't rely on group comparison). A better comparison would be to also use 8 rollouts in your algorithm. Would the algorithm overfit in that case?

The learning curves are expectedly noisy, I would recommend repeating the experiments instead of using exponential smoothing.

**Questions:**

Please see the weaknesses section. Could you further clarify why your algorithm works, in response to the weaknesses above?

Could you explain figure 6? I don't understand if the colors appearing on the same bar mean anything.

It may be beneficial to re-iterate what \pi_theta in line 296 means.

Could you provide a table to summarize the final accuraries (and other metrics) of models vs datasets vs training algorithms?

Are the log probabilities normalized by length?

---

### Note · Authors · 2025-11-26

**Comment:**

We thank the reviewers for their valuable feedback. After careful consideration, we have decided to withdraw this submission and will continue working to improve the quality of the paper based on reviewers' comments.

**Withdrawal Confirmation:**

I have read and agree with the venue's withdrawal policy on behalf of myself and my co-authors.